# Improving Fragment-Based Deep Molecular Generative Models

**Panukorn Taleongpong** [1]   **Brooks Paige** [1]

## Abstract

Deep molecular generative models have shown promising results and paved a new way for drug discovery. Their ability to explore the molecular space, estimated to be $10^{60}$, is significantly greater than traditional methods used for the virtual screening of existing databases. We introduce a novel fragmentation algorithm particularly suitable for use in deep generative models. In contrast to existing fragmentation algorithms, our procedure sequentially breaks a molecule using the Breaking of Retrosynthetically Interesting Chemical Substructure (BRICS) algorithm in such a manner that the linearization of fragments is directly invertible, guaranteed to be able to reconstruct the original molecule from the fragment sequence. This makes it appropriate for use in deep generative models trained with sequential models as likelihoods. We compare with previous fragment-based SMILES VAE methods and observe that our approach significantly enhances coverage of the molecular space and outperforms on distribution learning benchmarks.

## 1. Introduction

The process of drug discovery is lengthy, expensive and highly complex. Typically, the entire process to launch a drug takes, on average, 13.5 years (Paul et al., 2010) with a baseline capitalised cost of $2.6 billion (DiMasi et al., 2016). Traditional methods in drug discovery also suffer from high drug attrition rates - the failure rate of pharmaceutical development. Overall, many decisions in this process are made using expert knowledge, which is highly susceptible to bias. With a molecular space estimated to be over $10^{60}$ (Bohacek et al., 1996), these biases may result in an inefficiently explored space. Artificial Intelligence (AI) and Machine Learning (ML) methods have been increasingly

[1]Department of Computer Science, University College London, London, United Kingdom. Correspondence to: Panukorn Taleongpong <p.taleo17@gmail.com or ucabpta@ucl.ac.uk>, Brooks Paige <b.paige@ucl.ac.uk>.

*Accepted at the 1st Machine Learning for Life and Material Sciences Workshop at ICML 2024.* Copyright 2024 by the author(s).

popular tools used in the process of drug discovery as they overcome the challenges of traditional methods in drug discovery and can be applied to help at almost all stages of the drug discovery process (Kim et al., 2020). This paper focuses on improving current SMILES-based and Fragment-based Drug Discovery (FBDD) models, specifically deep generative models for distribution learning.

## 2. Literature Review

The most influential work that paved the way for research in generative models for *de novo* molecular design is by Gómez-Bombarelli et al. (2018), who proposed a character Variational Autoencoder (CVAE). This model encodes character-based discrete representations of molecules into continuous latent vectors. After which, a predictor estimates the chemical properties given the latent vectors, and the decoder converts the continuous vectors back into the character-based discrete representations of molecules. Bayesian Optimisation (BO) can then be used to generate molecules optimised with desired properties. This is a significant contribution to exploring the molecular space efficiently without needing prior knowledge of molecular construction. A major issue with SMILES-based models is they often generate pathological molecules which are difficult to synthesize, unstable, or even 'invalid' due to SMILES parse errors. Follow-up work has address this by incorporating higher-order structure in the data representation, rather than operating on pure text (Kusner et al., 2017; Jin et al., 2018; Bradshaw et al., 2019). Podda et al. (2020) tackles these issues by developing a fragment-based language model that generates a molecule fragment by fragment rather than atom by atom. This improves validity since fragments are already chemically valid. Additionally, infrequent fragments are masked with an identifier token to improve the uniqueness of molecules generated. During generation, if the masking token is sampled, a fragment is randomly sampled from the set of infrequent fragments.

## 3. Methodology

We improve upon the methodology developed by Podda et al. (2020) by introducing a more efficient fragmentation algorithm and using richer molecular embeddings. We refer to the model and methods by Podda et al. (2020) as the

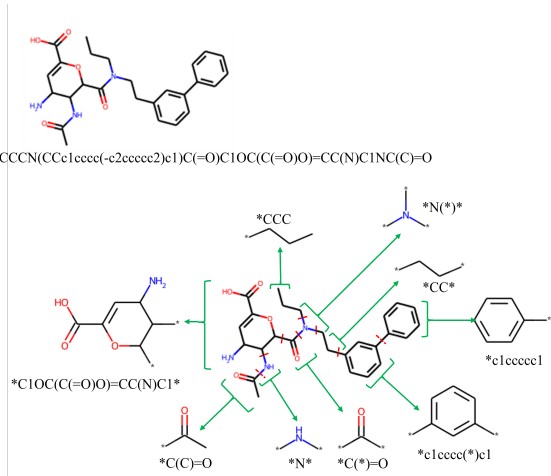

CCCN(CCc1cccc(-c2ccccc2)c1)C(=O)C1OC(C(=O)O)=CC(N)C1NC(C)=O

Figure 1: Example of fragments derived from a molecule using the BRICS algorithm. The asterisks represent dummy atoms, and the red dotted lines denote the 'cuts' in the molecule.

*benchmark*, with all experiments in this research undertaken on the ZINC dataset (Irwin et al., 2012).

### 3.1. Fragmentation

Our fragmentation process uses the same procedure as that performed by Podda et al. (2020) to identify breakable bonds: the Breaking of Retrosynthetically Interesting Chemical Substructure (BRICS) algorithm (Degen et al., 2008). The BRICS algorithm identifies strategic bonds based on medicinal chemistry concepts and performs retrosynthetic cuts simultaneously. Figure 1 shows an example of a molecule given by its SMILES representation 'CCCN(CCc1cccc(-c2ccccc2)c1)C(=O)C1OC(C(=O)O)=CC(N)C1NC(C)=O' broken down into fragments using the BRICS algorithm. These fragments consist of the following '*c1cccc(*)c1', '*CCC', '*C(C)=O', '*C1OC(C(=O)O)=CC(N)C1*', '*CC*', '*N(*)*', '*N*', '*C(*)=O' and '*c1ccccc1'. For our research, and based on the work of (Bowman et al., 2016), we view fragments as words that make up the molecule, which is the sentence. Hence, the fragmentation process must be more constrained to ensure that the fragments can be sequentially represented and that the original molecule can be reconstructed. Algorithm 1 details the pseudocode for our breadth-first fragmentation algorithm. Equivalent to the benchmark algorithm, our algorithm scans the original molecule in the order imposed by the canonicalised SMILES representation. When a BRIC bond is identified, the molecule is cut at that point with 'dummy' atoms attached to the end of the cleavage sites (Podda et al., 2020). Our algorithm improves upon the benchmark by ensuring that the molecule is cut to minimise

---

**Algorithm 1** Breadth First Fragmentation

---

**Input:** Original Molecule $M_0$, Fragment $F$, Fragment List $F_L$, Counter $C$, BRIC Bond Limit $B_{lim}$
Initialize $fragComplete = false$.
Initialize $F_L = []$
Initialize $B_{lim} = 0$
$F = M_0$
**function** Fragment($M_0$, $F$, $F_L$, $C$, $B_{lim}$):
  **if** $fragComplete = true$ **then**
    **return** $F_L$
  **end if**
  BRICBondsList = getBRICBonds($F$)
  **if** LEN(BRICBondsList) $< B_{lim}$ **then**
    $fragComplete = true$
    **return** $F_L$
  **else**
    **for** Bond $b$ **in** BRICBondsList **do**
      $F_{head}, F_{tail}$ = BreakOnBond($F$, $b$)
      **if** LEN(getBRICBonds($F_{head}$) $\leq B_{lim}$ **then**
        **if** checkReconstruction($F_L$, $F_{head}$, $F_{tail}$, $M_0$) $== true$ **then**
          append($F_L$, $F_{head}$)
          Fragment($M_0$, $F_{tail}$, $F_L$, $C$, $B_{lim} = 0$)
        **else if** LEN(BRICBondsList) $== 1$ **then**
          append($F_L$, $F$)
          $fragComplete = true$
        **else if** $b == BRICBondsList[-1]$ **then**
          Fragment($M_0$, $F_{tail}$, $F_L$, $C$, $B_{lim} + 1$)
        **end if**
      **else if** LEN(getBRICBonds($F_{tail}$) $\leq B_{lim}$ **then**
        Repeat with $F_{tail}$
      **end if**
    **end for**
  **end if**
**end function**

---

the number of BRIC bonds in the leaf fragment, allowing the procedure to be repeated on the internal fragment with more BRIC bonds. This relaxes the benchmark algorithm's key constraint of only allowing fragment extraction from left to right. Our algorithm produces a tree of fragments where the root node is the original molecule, and the leaves are the most simple fragments that make up the molecule. Although our fragmentation extraction process is no longer constrained in direction, we must store the fragments sequentially to represent them as words. We ensure that the original molecule can be reconstructed at every fragmentation step, meaning that the final sequence of fragments can be recombined from right to left to form the original molecule. Algorithm 2 details the fragment reconstruction process. Using the same molecule example as in Figure 1, Figure 2 and Table 1 detail the fragments produced using our breadth-first recursive fragmentation

**Algorithm 2** Fragment Reconstruction

**Input:** Original Molecule $M_0$, Fragment List $F_L$
  $F = M_0$
  **function** checkReconstruction($M_0$, $F_L$:
    $F_{leaf} = F_L[-1]$
    **for** i **in** LEN($F_L$)-1 **do**
      $F_{internal} = F_L[-i-2]$
      $F_{rec}$ = replaceLastDummy($F_{leaf}$, $F_{internal}$)
    **end for**
    **if** canonicalise($F_{rec}$) == $M_0$ **then**
      **return** True
    **else**
      **return** False
    **end if**
  **end function**

| TYPE | FRAGMENT |
|---|---|
| **HEAD FRAGMENT** | **\*CCC** |
| RECURSE TAIL | N(\*)(CCc1cccc(       -c2ccccc2)c1)C(=O)C1OC(C(=O)O)=CC(N)C1NC(C)=O |
| **TAIL FRAGMENT** | **\*CCc1cccc(-c2ccccc2)c1** |
| RECURSE HEAD | N(\*)(\*)C(=O)C1OC(C(=O)O)=CC(N)C1NC(C)=O |
| **HEAD FRAGMENT** | **\*N(\*)\*** |
| RECURSE TAIL | C(\*)(=O)C1OC(C(=O)O)=CC(N)C1NC(C)=O |
| **HEAD FRAGMENT** | **\*C(\*)=O** |
| RECURSE TAIL | C1(\*)OC(C(=O)O)=CC(N)C1NC(C)=O |
| **HEAD FRAGMENT** | **\*NC1C(N)C=C(C(=O)O)OC1\*** |
| RECURSE TAIL | C(\*)(C)=O |
| **FINAL FRAGMENT** | **C(\*)(C)=O** |

Table 1: Fragments and Resulting Molecule

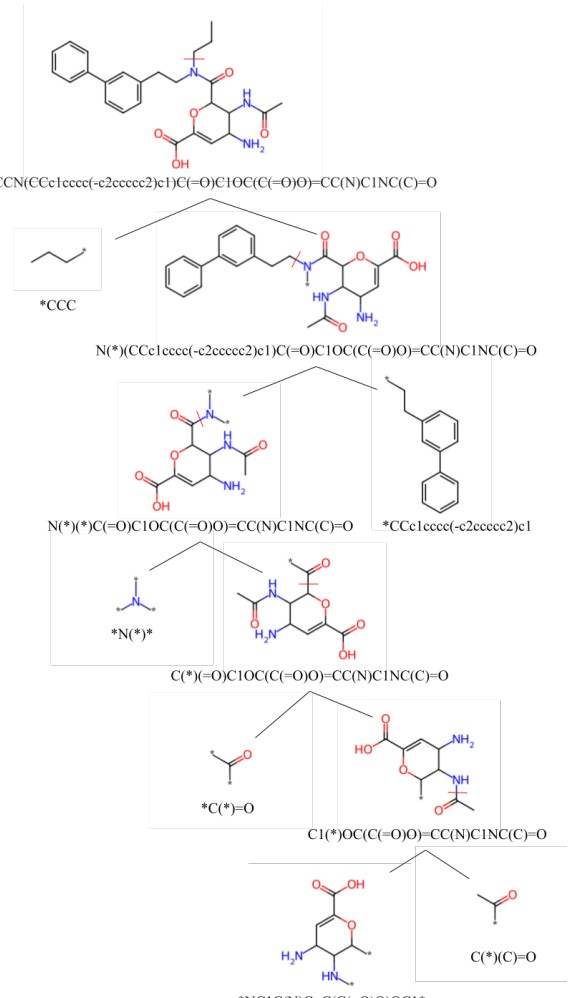

Figure 2: Example of fragments derived from a molecule using Breadth-First Fragmentation algorithm. The asterisks represent dummy atoms, and the red dotted lines denote the 'cuts' in each recursion.

algorithm. 'Head' and 'Tail' fragments are the leaf fragments concatenated into the final set of fragments in the order they are produced.

Setting the limit of atoms per fragment to one for both the benchmark and our fragmentation algorithm, Figures 3 and 4 clearly show that our algorithm is more efficient. The efficiency is calculated as:

$$\text{Fragmentation Efficiency} = \frac{\text{Number of fragments}}{\text{Number of BRICS Bonds + 1}}$$

Our algorithm has a mean fragmentation efficiency of 79% compared to the benchmark, 47%. Discounting molecules that cannot be fragmented, the number of unique fragments produced with our algorithm is 93,557, and for the benchmark algorithm, it is 186,726, reflecting respective vocabulary sizes. This is significantly larger than the vocabulary of other molecular generative models, such as CVAE (35). This poses the significant issue of infrequent fragments, which, at sampling time, will have extremely low probabilities. The benchmark method alleviates this issue by using low-frequency masking and sampling. On the other hand, our fragmentation algorithm addresses this problem directly by producing a significantly lower number of unique fragments. Notably, our algorithm producing fewer unique fragments also means that more than twice (74%) the number of molecules in a randomised held-out test set of 20,000 molecules can be reconstructed compared to the benchmark algorithm (34%).

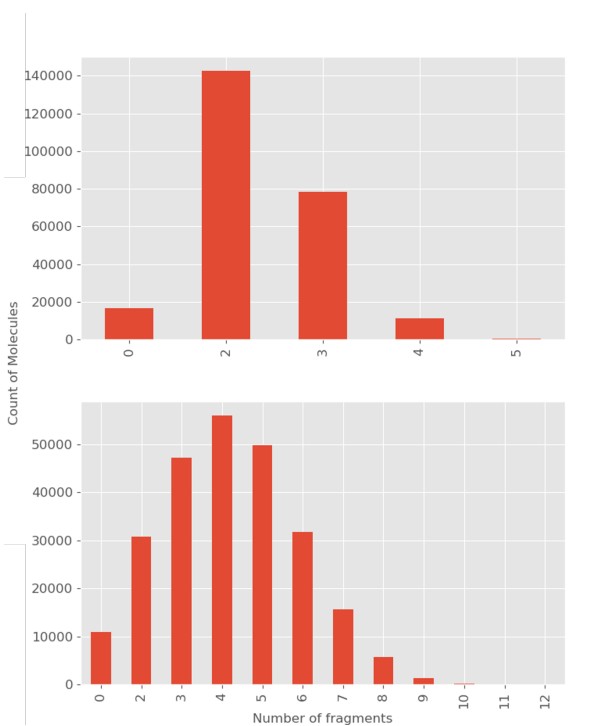

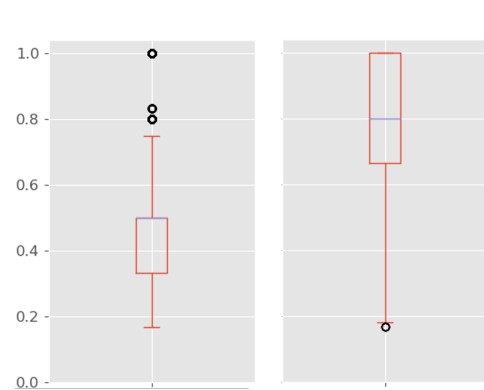

Figure 4: Fragment efficiency using Podda et al. (2020) (left) and our method (Right).

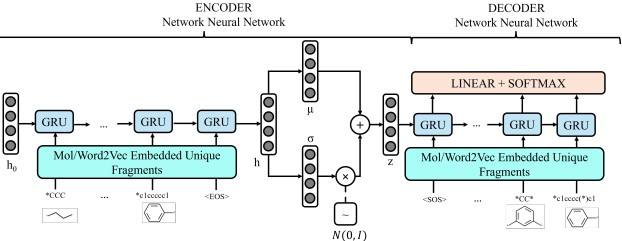

Figure 5: Our model.

Figure 3: Fragment count histogram using Podda et al. (2020) (Top) and our method (Bottom).

### 3.2. Fragment Embedding

By representing a sequence of fragments as $s = (s_1, s_2, ..., s_{|s|})$, where $s_i$ represents each fragment in a SMILES string, Podda et al. (2020) trained the embeddings using a skip-gram Word2Vec model (Mikolov et al., 2013) with negative sampling (Le & Mikolov, 2014).

For our research, we utilise and compare both the skip-gram Word2Vec model and pre-trained skip-gram Mol2Vec model (Jaeger et al., 2018) for fragment embedding. The benefit of the pre-trained Mol2Vec model is that it has been trained using over 20 million compounds (ZINC database). Hence, the molecule substructure embeddings are incredibly rich. Furthermore, the embeddings can simply be summed to produce a rich embedding of the whole molecule (Jaeger et al., 2018). This is particularly beneficial in our model architecture, which combines fragments sequentially, ensuring a rich representation of the molecules encoded.

### 3.3. Training

Our model consists of an encoder and decoder network as detailed in Figure 5. We utilise Kullback-Leibler (KL) annealing to prevent posterior collapse and found that a scheduling weight parameter value of $\beta = 1 \times 10^{-6}$ is the most effective in doing so. Our model was trained end-to-end on the training set $\mathcal{D}$. The overall loss of the

model, $\mathcal{L}(\mathcal{D})$ is the sum of the encoder loss, $\mathcal{L}_{\text{enc}}(x)$, and the decoder loss, $\mathcal{L}_{\text{dec}}(x)$:

$$\mathcal{L}(\mathcal{D}) = \sum_{x \in \mathcal{D}} \left( \beta \mathcal{L}_{\text{enc}}(x) + \mathcal{L}_{\text{dec}}(x) \right) \qquad (1)$$

Note that the training set used limits the atoms per fragment to three, which is the same constraint as the benchmark model.

### 3.4. Sampling

As our scheduling weight parameter is extremely small, the latent space is not expected to be effectively represented with $N(0, I)$. Therefore, we estimated the latent mean $\bar{z}$ and standard deviation $s_z$ by encoding a sample of the training set into the latent space using the trained encoder. We then start the sampling process by sampling a latent vector from the latent space $z \sim \mathcal{N}(\bar{z}, s_z^2)$. This is the initial state of the decoder. An SoS token is inputted into the decoder, producing an output probability over all tokens through a softmax layer. The next token is selected greedily and used as the input for the next decoding step. This process repeats until an EoS token is sampled or the maximum sampling length is reached. The molecule is then constructed by combining the sampled fragments using our reconstruction algorithm (Algorithm 2). We constrain the sampling process to terminate only if the final molecule produced is valid. To

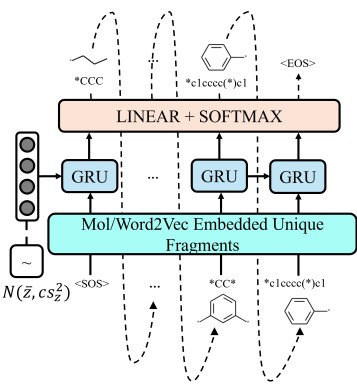

Figure 6: Our sampling procedure.

| MODEL | VALID | NOVEL | UNIQUE | SAMPLE RATE (MS/MOL) |
|---|---|---|---|---|
| **WORD2VEC EMBEDDING** | | | | |
| PODDA | 1 | 0.992 | 0.460 | |
| OURS (C = 10) | 1 | 0.999 | 0.106 | 69 |
| OURS (C = 15) | 1 | 0.998 | 0.325 | 87 |
| OURS (C = 20) | 1 | 0.997 | 0.544 | 112 |
| **OURS (C = 25)** | **1** | **0.997** | **0.701** | **162** |
| **MOL2VEC EMBEDDING** | | | | |
| OURS (C = 10) | 1 | 0.999 | 0.144 | 58 |
| OURS (C = 15) | 1 | 0.997 | 0.423 | 79 |
| OURS (C = 20) | 1 | 0.997 | 0.661 | 123 |
| **OURS (C = 25)** | **1** | **0.997** | **0.793** | **211** |
| **LOW FREQUENCY MASK SAMPLING** | | | | |
| PODDA (LFM) | 1 | 0.995 | 0.998 | |

Table 2: Results of 20,000 molecules sampled

improve the uniqueness of molecules generated, we apply a factor, $c$, to the variance when sampling the latent vector ($z \sim \mathcal{N}(\bar{z}, c \times s_z^2)$). Figure 6 portrays the general sampling procedure.

## 4. Results

The results from sampling 20,000 molecules from our model using Word2Vec and Mol2Vec embeddings compared to those reported by Podda et al. (2020) are detailed in table 2. Furthermore, figures 7 and 8 portray the summary statistics of the properties of our generated molecules compared to ZINC. It is clear that our models, regardless of the embedding method, outperform the benchmark significantly in uniqueness without having to rely on low-frequency mask sampling. Our models also demonstrate slight improvement in novelty compared to the benchmark with and without low-frequency mask sampling. Notably, our model with Mol2Vec embedding outperforms that with Word2Vec embedding. This may be indicative of a more expressive latent space when Mol2Vec is used. We also note that improving uniqueness by simply increasing the factor applied to the sampling variance comes at a cost to the sampling efficiency.

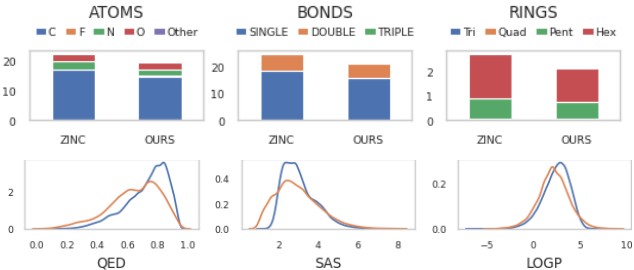

Figure 7: Top: Statistics of sampled molecular characteristics. Bottom: Density of sampled molecular properties where the blue line corresponds to ZINC. Our model: c = 25, Mol2Vec Embedding.

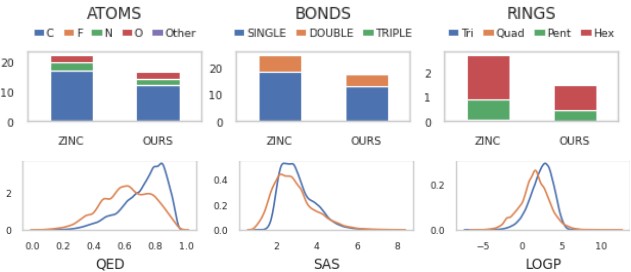

Figure 8: Top: Statistics of sampled molecular characteristics. Bottom: Density of sampled molecular properties where the blue line corresponds to ZINC. Our model: c = 25, Word2Vec Embedding.

## 5. Conclusion

Our results demonstrate advancements in SMILES-based fragment-based drug discovery models for distribution learning in the following ways:

1. We propose a novel fragmentation and fragment reconstruction algorithm that produces fragments much more efficiently whilst guaranteeing the reconstruction of the original molecule.

2. Our model demonstrates superior performance metrics of the generated molecules by simply applying a factor to the sampling variance rather than using low-frequency masking.

3. Utilisation of fingerprint-based Mol2Vec fragment encoding improves distribution learning results and paves the way for SMILES-based fragment-based property prediction and molecular optimisation.

The next steps to extend this model include extending from a distribution-learning model to a goal-directed model, and investigating different methods to prevent posterior collapse, such as average and max pooling (Long et al., 2020).

## Software and Data

For reproducibility, we publicly release the code repository for this project at https://github.com/panukorn17/DEFRAGMO.

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

# Appendix

## Molecular Representation

Molecular representation is a critical consideration in drug discovery. It constrains what models can be used and has distinct advantages and limitations. This section details two popular molecular representations used in this paper: SMILES and molecular fingerprints.

### SMILES

The Simplified Molecular Input Line Entry System (SMILES) represents a molecular graph as a sequence of atoms and bonds using short ASCII strings. The atoms in SMILES strings are represented by the abbreviations of their elements. The key advantage of SMILES representation of molecules is that it is highly interpretable, allowing deep learning and language models to be easily applied. However, a significant disadvantage is that they lack invariance. Similar molecules can be represented by significantly different SMILES strings. Furthermore, the same SMILES string may also have many valid forms due to molecule conformation. For example, CCO, OCC, and C(O)C are the same molecule as portrayed in Figure 9. Canonicalisation is used to solve this issue. This process transforms the string into its graphical molecular form and then produces a SMILES string that is unique for the molecule.

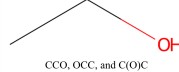
CCO, OCC, and C(O)C

Figure 9: Molecular structure of CCO, OCC, and C(O)C

### MOLECULAR FINGERPRINTS

Molecular fingerprints are chemical structure representations originally developed to aid chemical substructure searching (Todeschini & Consonni, 2000). Rogers & Hahn (2010) extended this by introducing Extended-Connectivity Fingerprints designed to capture molecular features relevant to molecular activity. This is done by assigning identifiers to the atoms in a molecule, concatenating neighbouring atom identifiers, applying a hash function to represent identifier arrays as integers and adding this to the set of fingerprints. Figure 10 demonstrates this algorithm. This representation

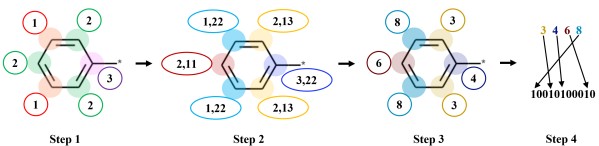

Figure 10: Example molecular fingerprint process

is highly popular due to its superior performance in the virtual screening (Riniker & Landrum, 2013) and molecular activity prediction (Mayr et al., 2016; Awale & Reymond, 2019) tasks. It is also highly interpretable and simple to apply using RDKit (Landrum, 2012). However, key disadvantages of this algorithm include its high dependence on initial conditions, the fact that similar substructures can be mapped onto different bits, sparseness, and hashing collisions.

## Fragmentation

Figures 11 and 12 show the most frequent fragments produced by Podda et al. (2020) and our fragmentation algorithms, respectively, when constrained to having at least one atom per fragment. Since our models are trained on fragments generated by our algorithm when constrained to having at least three atoms per fragment, we reproduce the fragment count histogram and fragment efficiency under these conditions in Figures 13 and 14, respectively.

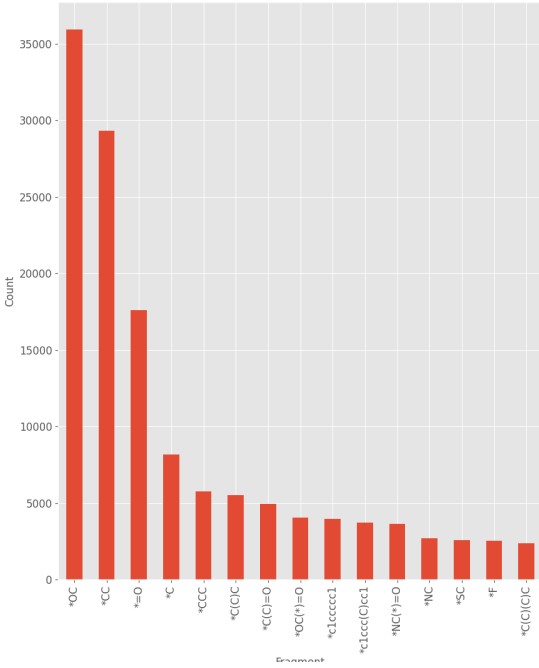

Figure 11: Most frequent fragments with Podda's method.

## Training

### EMBEDDING

The process of embedding molecules using Mol2Vec consists of two steps: constructing a molecular sentence with a specified Morgan fingerprint radius and generating the

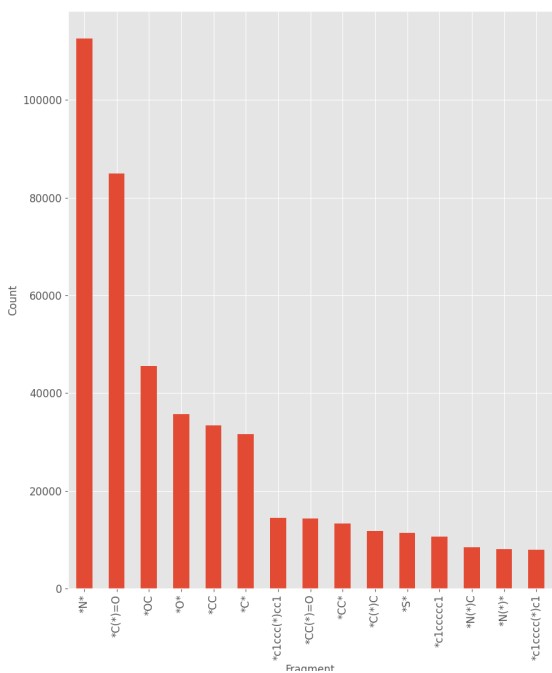

Figure 12: Most frequent fragments with our method.

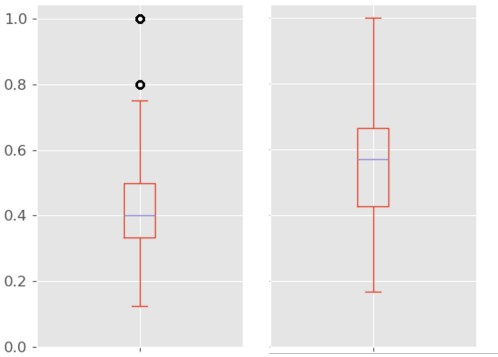

Figure 13: Fragment efficiency using Podda et al. (2020) (left) and our method (Right).

| HYPER-PARAMETER | WORD2VEC (BENCHMARK MODEL) | PRE-TRAINED MOL2VEC (OUR MODEL) |
|---|---|---|
| EMBEDDING SIZE | 100 | 100 |
| EMBEDDING WINDOW SIZE | 3 | 10 |
| DATA | THE TRAINING SET | ZINC |
| MORGAN FINGERPRINT RADIUS | - | 1 |

Table 3: The hyper-parameters used to train each embedding model

molecular embedding from the pre-trained Mol2Vec model. The key difference between using the skip-gram Word2Vec model (benchmark) and the skip-gram Mol2Vec model (ours) is that the former considers an individual fragment as a one-hot-encoded vector before training the model to attain the embedding of that fragment. The latter considers the molecule (or, in our case, fragment) substructures through the Morgan Algorithm (Rogers & Hahn, 2010) before training the model to produce the embedding of the substructure, which is then summed to generate the embedding of the entire fragment. Figure 15 and 16 portray the architectures used to train the fragment embeddings for the benchmark and our models, respectively. Figure 17 portrays the architecture used to look up fragment embeddings with Mol2Vec. Table 3 details the hyper-parameters used to train each embedding model.

## MODEL

Our model's encoder utilises Gated Recurrent Units (GRUs) (Cho et al., 2014). Fragment embeddings $x_i$ are processed to hidden representations $h_i$ through the following process

via the GRUs:

$$r_i = \text{sigmoid}(W_r x_i + U_r h_{i-1} + b_r) \quad (2)$$
$$u_i = \text{sigmoid}(W_u x_i + U_u h_{i-1} + b_u) \quad (3)$$
$$\hat{h}_i = \tanh(W_h x_i + U_h(r_i \odot h_{i-1}) + b_h) \quad (4)$$
$$h_i = u_i \odot \hat{h}_i + (1 - u_i) \odot h_{i-1} \quad (5)$$

- $r_i$, $u_i$, $\hat{h}_i$ and $h_i$ are the reset gate, update gate, candidate activation, and output vectors of the $i^{th}$ fragment, respectively.

- $W$, $U$, and $b$ are the learned weights matrices and biases.

- $\odot$ is the element-wise multiplication operation.

The encoder is trained to minimise the KL divergence:

$$\mathcal{L}_{\text{enc}}(x) = -\text{KL}(\mathcal{N}(\mu, \sigma^2 I)||\mathcal{N}(0, I)) \quad (6)$$

- $\mu$ and $\log(\sigma^2)$ is learnt through a one-layer neural net i.e. $\mu = W_\mu h + b_\mu$ and $\log(\sigma^2) = W_\sigma h + b_\sigma$.

- $W$ is the learned weight matrix, $b$ is the learned bias vector, and $h$ is the final output vector from the GRUs.

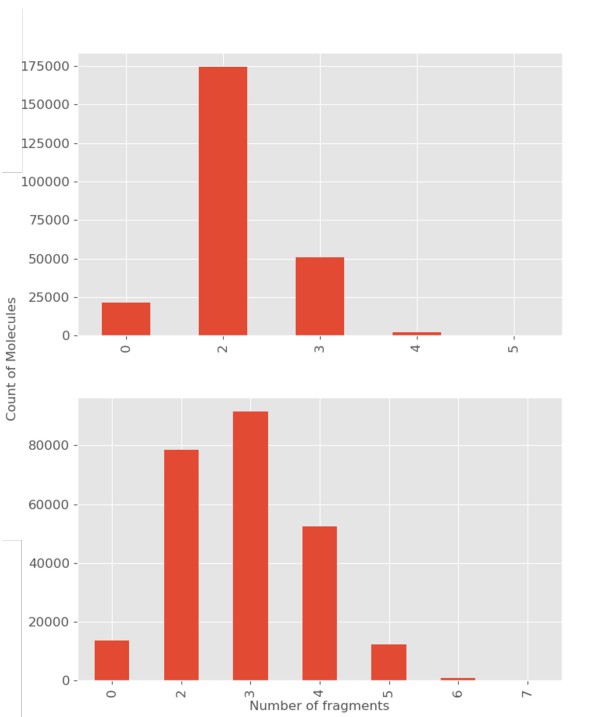

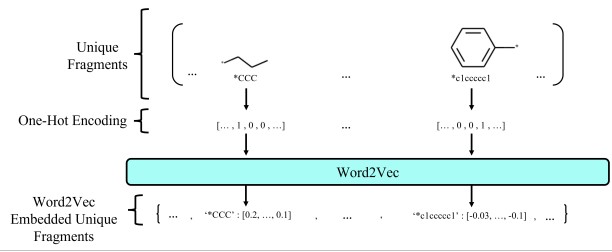

Figure 14: Fragment count histogram using Podda et al. (2020) (Top) and our method (Bottom).

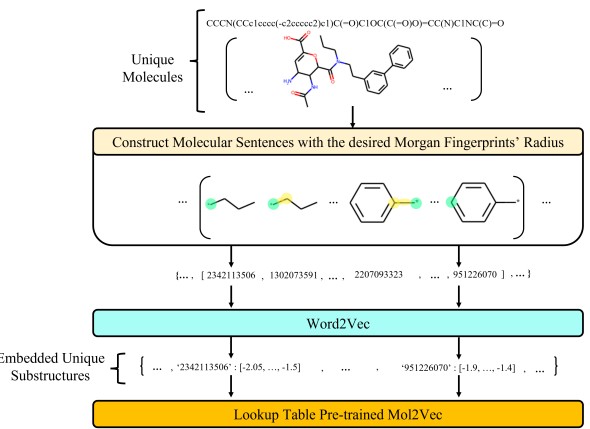

Figure 16: Mol2Vec architecture used to train the unique molecule substructure embeddings

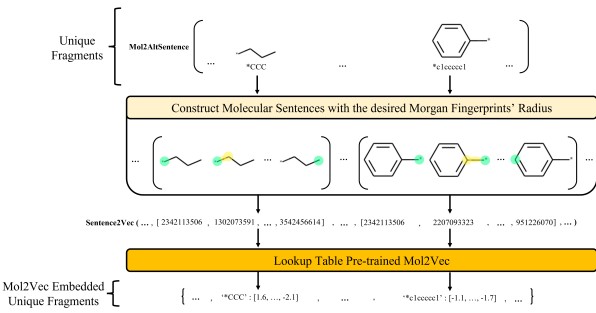

Figure 17: Architecture used to lookup the fragment embeddings for our model

Figure 15: Word2Vec architecture used to train the fragment embeddings for the benchmark model

The decoder is a recurrent model with GRU units that produce the probability of the next fragment given the current fragment in the sequence. Using the reparameterisation trick (Kingma & Welling, 2014), the hidden stage is initialised as:

$$z = h_0 = \mu + \sigma\epsilon, \text{ where } \epsilon \sim \mathcal{N}(0, I) \tag{7}$$

The decoder utilises teacher forcing during training (Williams & Zipser, 1989) to provide the correct next fragment rather than the model sampling from the fragment distribution. This forces the RNN to remain close to the ground-truth fragment sequence (Lamb et al., 2016). The probability of the next fragment is then computed as follows:

$$P(x_{i+1}|x_i, h_{i-1}) = \text{softmax}(W_{out}h_i + b_{out}) \tag{8}$$

where $h_i = \text{GRU}(x_i, h_{i-1})$. With teacher enforcing, the decoder is trained to minimise the cross-entropy loss between the correct sequence of fragments and their predicted probabilities as follows:

$$\mathcal{L}_{\text{dec}}(x) = -\sum_{i=1}^{|x|} \log P(x_{i+1}|x_i, h_{i-1}) \tag{9}$$

The final hyper-parameters used to train our model are detailed in Table 4.

KL ANNEALING

KL annealing (Bowman et al., 2016) is used to overcome KL vanishing, symptomatic of posterior collapse (Bowman et al., 2016; Yang et al., 2017; Higgins et al., 2017). This is particularly problematic for RNN-based VAE models (He et al., 2022). Posterior collapse occurs during training when the model falls into the local optimum of the ELBO objective and the variational posterior $q_\phi(\mathbf{z}|\mathbf{x})$ mimics the model prior $p(\mathbf{z})$ leading to the decoder ignoring the latent vectors. KL annealing is applied through a scheduled weight

| Hyper-Parameter | Benchmark Value | Our Value |
|---|---|---|
| Epochs | 4 | 4 |
| Batch Size | 128 | 128 |
| Hidden Layers | 2 | 2 |
| Hidden Size | 128 | 128 |
| Latent Size | 100 | 100 |
| Dropout | 0.3 | 0.3 |
| Embedding Size | 64 | 100 |
| Learning Rate | 1E-05 | 1E-05 |
| KL Annealing ($\beta$) | 0.9 | 1E-06 |

Table 4: Hyper-parameters used to train the (Podda et al., 2020) and our models

parameter $\beta \in [0, 1]$ on the KL term in the loss function:

$$\mathcal{L}_{\text{enc}}(x) = \beta \times -\text{KL}(\mathcal{N}(\mu, \sigma^2 I)||\mathcal{N}(0, I)) \qquad (10)$$