# OpenReview forum: "Improving Fragment-Based Deep Molecular Generative Models"
_ICML.cc/2024/Workshop/ML4LMS — ML4LMS Oral_

### Official Review · Reviewer_yH1w · 2024-06-05
**Well presented work**

**Rating:** 7
**Confidence:** 3

**Review:**

Pros:
- Clearly written, well presented work
- Novel fragmentation algorithm

Cons:
- Comparison to other SOTA generative models would have been welcome (incl non fragment based) for context of the significance of the submitted work
- Performance of generative model for an objective function compared to the benchmark model would also have been nice to see.

---

### Official Review · Reviewer_viNe · 2024-06-10
**Well-written paper presenting a strong algorithmic approach, but confusing results which warrants better explanation of experimental results and baselines.**

**Rating:** 7
**Confidence:** 2

**Review:**

**Summary:** this paper introduces a novel fragmentation algorithm for use in deep generative models of molecules trained with sequential models, which significantly enhances coverage of the molecular space and outperforms on distribution learning benchmarks.

**Strengths:**
- Well written paper detailing an interesting algorithm and presenting clear and significant results in molecular generation.
- The paper introduces succinct and clear motivation for the work.
- Originality of the approach is very clear, and the fragmentation algorithm is well presented.
- Contributions to the field are clear and interesting.

 **Weaknesses:**
-  The presentation of the results in section 4, table 2 and figures 7 and 8 is confusing, making a fair review of experimental validity and the significance of the contribution difficult.

**Questions & Limitations:**
- **Fragmentation efficiency:** what does a higher fragmentation efficiency imply for the quality of the models trained using this method? It would be helpful to discuss this and why the improvement in efficiency is experimentally meaningful.
- **Novelty vs uniqueness trade-off:** from table 2 it appears the small increase in novetly from Podda's LFM baseline is at the expense of uniqueness which is significantly lower in the model presented here. It would be helpful to discuss this.
- **Explaining results:** contrary to what appears in table 2, section 4 (Results) claims *"It is clear that our models, regardless of the embedding method, outperform the benchmark significantly in uniqueness without having to rely on low-frequency mask sampling"*. This isn't clear to this reviewer--please provide a detailed explanation, check that table 2 is accurate, or explicitly detail the benchmarks used for figures 7 and 8 if they differ.

**Minor typos/edits**
- Line 20 left column: explicit BRIC bonds (meaning is not common knowledge).
- Figure 1 may be better than line 86 to 92 to present the SMILES of the molecule and its fragments, if you increase font size.

---

### Official Review · Reviewer_8hPC · 2024-06-11
**The manuscript "Improving Fragment-Based Deep Molecular Generative Models" advances deep molecular generative models for drug discovery by introducing an efficient, invertible fragmentation algorithm. This method enhances model training and molecular space exploration.**

**Rating:** 9
**Confidence:** 3

**Review:**

The research is of high quality, presenting methodological improvements backed by empirical evidence using the ZINC dataset. The paper is clearly written and well-structured, with technical methodologies effectively explained and illustrated with appropriate visuals and algorithmic descriptions. The work is original, featuring a new fragmentation method that enhances the functionality of generative models used in drug discovery. Its significant potential impact lies in its ability to reduce both the time and cost associated with drug development, addressing a critical need within the pharmaceutical industry.

The study's primary strengths include its innovative approach to molecule fragmentation that increases efficiency and the capability to generate more diverse molecular structures. It also convincingly demonstrates performance improvements over existing models through extensive testing.